# The Role of Venous Blood Gas Analysis in Critical Care: A Narrative Review

**DOI:** 10.3390/medicina61081337

**Published:** 2025-07-24

**Authors:** Dario Giani, Michele Cosimo Santoro, Maurizio Gabrielli, Roberta Di Luca, Martina Malaspina, Maria Lumare, Licia Antonella Scatà, Martina Pala, Alberto Manno, Marcello Candelli, Marcello Covino, Antonio Gasbarrini, Francesco Franceschi

**Affiliations:** 1Department of Medical and Surgical Sciences, Fondazione Policlinico Universitario A. Gemelli IRCCS, Università Cattolica del Sacro Cuore, 00168 Rome, Italy; dario.giani01@icatt.it (D.G.); maurizio.gabrielli@policlinicogemelli.it (M.G.); antonio.gasbarrini@unicatt.it (A.G.); 2Department of Emergency, Anesthesiological and Reanimation Sciences, Fondazione Policlinico Universitario A. Gemelli IRCCS, Università Cattolica del Sacro Cuore, 00168 Rome, Italy; roberta.diluca@policlinicogemelli.it (R.D.L.); malaspinamartina@gmail.com (M.M.); lumaremaria@gmail.com (M.L.); licia141292@gmail.com (L.A.S.); martinapala319@gmail.com (M.P.); alberto.manno@policlinicogemelli.it (A.M.); marcello.candelli@policlinicogemelli.it (M.C.); marcello.covino@policlinicogemelli.it (M.C.); francesco.franceschi@unicatt.it (F.F.)

**Keywords:** blood gas analysis, arterial, venous, emergency department, critical care

## Abstract

ABG analysis is the gold standard for assessing acid–base balance, oxygenation, and ventilation in critically ill patients, but it is invasive and associated with patient discomfort and potential complications. Venous blood gas (VBG) analysis offers a less invasive alternative, although its clinical utility remains debated. This review evaluates the current evidence on VBG analysis, exploring its correlation with ABG, clinical applications, and limitations. Studies show a strong correlation between ABG and VBG for pH and a good correlation for bicarbonate and base excess in most cases, while the correlation for pCO_2_ remains controversial. Predictably, pO_2_ values differ significantly due to oxygen consumption gradients between the arterial and venous blood. VBG analysis is especially valuable for initial assessments, monitoring therapeutic responses, and guiding resuscitation in intensive care settings. It is not merely an alternative to ABG but a complementary tool that can provide unique insights, such as mixed venous oxygen saturation (SvO_2_) or indices that require combined ABG and VBG data, like the pCO_2_ gap. This review highlights the diagnostic equivalence of VBG in appropriate contexts and advocates for its use when arterial sampling is unnecessary or impractical. Furthermore, VBG analysis could enhance patient care by enabling the timely, less invasive assessment of hemodynamic and metabolic conditions. Future research should focus on refining interpretation algorithms and expanding the clinical applications of VBG to fully realize its potential in critical care practice.

## 1. Introduction

ABG analysis is currently considered the gold standard for measuring blood gases O_2_, and CO_2_. It is the test of choice for assessing oxygenation, ventilation, and acid–base status in acute or critically ill patients. Consequently, point-of-care blood gas analyzers have become a standard tool in critical care settings, particularly in the Emergency Department [1,2]. However, obtaining an arterial blood sample requires a procedure that is often painful for the patient and carries potential complications, such as thrombosis, ischemia, rupture, pseudoaneurysm, as well as the risk of needlestick injury to the operator. VBG analysis offers a less invasive alternative, though its clinical utility remains debated.

The aim of this review is to evaluate the current evidence on the utility of VBG, assessing its correlation with ABG, clinical applications, and limitations. An in-depth discussion of the pathophysiological basis of the presented data is also provided in order to support the use of VBG as a practical tool, not only as an alternative to arterial blood, but also as a complement to it in specific clinical settings.

## 2. The Literature Research

We conducted a review of the medical literature from the past 10 years to identify studies and reports on the clinical use of mixed, central, and peripheral venous blood in critically ill patients. The following electronic databases were searched: PubMed^®^, MEDLINE^®^, Scopus^®^, and the Cochrane Library^®^, using the following search terms:‘Venous blood gas analysis’ AND ‘Emergency Department’;‘Venous blood gas analysis’ AND ‘Critical Care’;‘Venous blood gas analysis’ AND ‘ICU (Intensive Care Unit)’.

This search yielded English-language articles, and it was primarily focused on reviews, systematic reviews, meta-analyses, abstracts, and clinical trials. Article selection was conducted independently by two reviewers (MCS and DG). Additionally, a manual search of the reference lists of the selected articles was performed to identify other potentially relevant studies. The results of this literature review are summarized in Table 1.

We did not assess the quality of the studies using validated tools due to the narrative nature of this review, which explores various aspects of the potential clinical utility of venous blood. In addition to systematic reviews and meta-analyses, we prioritized clinical studies with robust methodological designs, large sample sizes, and, when possible, more recent publication dates.

## 3. Differences in the Main Gas Analysis Parameters of Arterial and Venous Blood

Clinicians utilizing venous blood must be aware of the differences between arterial and venous blood gas values. Firstly, unlike arterial blood sampling, the site of venous sampling (e.g., peripheral or central venous catheter) significantly influences blood gas parameters—particularly oxygen and carbon dioxide values—which are affected by the local metabolic activity. Secondly, venous blood gas analysis does not simply serve as a substitute for arterial analysis but can also provide complementary information.

Under physiological conditions, electrolyte and hemoglobin levels are generally consistent between arterial and venous samples. Although pH and HCO_3_^−^ values differ slightly, they remain largely comparable: venous blood typically has a pH approximately 0.04 units lower and a bicarbonate level about 1 mEq/L higher than arterial blood [3]. In contrast, carbon dioxide and especially oxygen values differ substantially between arterial and venous blood (see Table 2).

PH tends to show a more consistent agreement between arterial and venous blood, whereas bicarbonate demonstrates slightly less concordance. Combined with the heterogeneity of patient populations and sampling types (i.e., mixed, central, and peripheral venous blood), this may help explain why many studies report a lack of agreement between arterial and venous pCO_2_ values. Intuitively, based on the Henderson–Hasselbalch equation, when pH and bicarbonate show enough concordance, one would expect a corresponding agreement in pCO_2_. However, as previously noted, this is not consistently observed in the literature. Table 3 summarizes the differences in blood gas values between arterial and mixed, central, and peripheral venous samples in normal conditions [3,4,5].

So, in conclusion, these values are similar but not identical. In the following chapters, it is explained how to interpret them and how to use them in clinical practice.

## 4. Mixed, Central, and Peripheral Venous Blood

### 4.1. Central Mixed Venous Blood

As previously mentioned, the site of venous blood sampling significantly influences blood gas values, particularly pO_2_ and pCO_2_, due to their susceptibility to local metabolic activity. Among the different types of venous samples, mixed venous blood is considered the most representative of the entire body, as it results from the combination of blood returning from both the superior and inferior vena cava. It is collected via a catheter positioned in the pulmonary artery (Swan-Ganz catheter), where blood flows from the right ventricle. The normal reference values for blood gas analysis performed on mixed venous blood are shown in Table 3.

Given these characteristics, mixed venous blood is commonly used for venous oximetry, i.e., the measurement of SvO_2_, where “v” denotes venous. SvO_2_ is considered a surrogate marker of CO. Under normal physiological conditions, arterial oxygen saturation is approximately 100%, while mixed venous saturation is around 75% [6,7]. The remaining 25% reflects the amount of oxygen extracted by the tissues. In pathological states with hemodynamic instability, tissue oxygen extraction increases, resulting in lower SvO_2_, typically falling below 70–75%, so it can be used as a marker of cardiac output, as described in Section 5.1.

In intensive care settings where a pulmonary artery catheter is available, SvO_2_ is a widely used parameter to assess hemodynamic instability in conditions such as sepsis, heart failure, and the perioperative and postoperative periods of major surgery, particularly cardiac surgery. However, due to the declining use of pulmonary artery catheters in clinical practice, ScvO_2_ is now more commonly measured using central venous blood.

The mean differences between arterial and venous values for all studies included in the Byrne’s meta-analysis [8], along with their 95% confidence intervals, are summarized in Table 2. The mean arterial pH across studies ranged from 7.15 to 7.46, while mean venous pH values ranged from 7.10 to 7.43. The estimated mean difference between venous and arterial pH was 0.033, with venous values being lower and the difference statistically significant [9].

Regarding pCO_2_ and pO_2_, the results revealed a wide range of values. Mean arterial pCO_2_ values ranged from 29.6 to 75.9 mmHg, while the mean venous pCO_2_ values ranged from 34.9 to 82.5 mmHg. The estimated bias was −4.15 mmHg. Heterogeneity was substantial, indicating that most of the variability was due to differences between studies. A venous pCO_2_ value of 55 mmHg could correspond to an arterial pCO_2_ between 44.3 mmHg (within the normal range) and 57.4 mmHg (consistent with type 2 respiratory failure).

A random-effects meta-analysis further demonstrated wide variability in pO_2_ values. Mean arterial pO_2_ ranged from 55.4 to 146.5 mmHg, while mean venous pO_2_ values ranged from 33.1 to 107.1 mmHg. The differences between arterial and venous pO_2_ were substantial across all studies [5,10].

### 4.2. Central Venous Blood

Central venous blood is obtained via a CVC, which is inserted into a large-caliber vein leading to the right atrium—commonly the internal jugular, subclavian, or femoral vein. Compared to the Swan-Ganz catheter used to measure SmvO_2_, the placement of a CVC is a less invasive method for assessing venous oxygenation. The oxygen saturation measured from central venous blood is referred to as ScvO_2_ and it is a marker of cardiac output, as reported below, in Section 5.1.

In clinical practice, ScvO_2_ is often used as a surrogate for SvO_2_. Although blood gas values from central venous blood do not perfectly match those from mixed venous blood, several studies have demonstrated a reasonable agreement, with ScvO_2_ typically differing from SvO_2_ by approximately 5–10% (Table 3).

As noted by Janotka et al., under physiological conditions, SvO_2_ tends to be slightly higher (typically 75–70%) than ScvO_2_ (~70%). However, this relationship may reverse in critically ill patients with hemodynamic instability due to the preferential redistribution of blood flow to vital organs, such as the brain and heart, at the expense of peripheral tissues, a phenomenon known as “circulatory centralization” [6]. In such cases, differences of up to 18% between ScvO_2_ and SvO_2_ have been reported [6].

It is not recommended to assess ScvO_2_ using a CVC inserted via the femoral vein, as this accesses the inferior vena cava. The resulting measurement (SfvO_2_) has been shown to differ significantly from values obtained from subclavian or jugular venous blood. In 2015, Martí et al. demonstrated that, unlike lactate, SfvO_2_ does not correlate well with ScvO_2_ or SvO_2_ [7], and these findings have since been corroborated by other studies [11,12]. For example, after intubation, venous oxygen saturation measured from the internal jugular or subclavian veins typically rises significantly, whereas SfvO_2_ remains unchanged. This discrepancy is largely attributed to circulatory centralization during cardiovascular compromise, wherein vasoconstriction reduces blood flow to splanchnic and peripheral regions. As perfusion to organs like the brain and heart is prioritized, their oxygen extraction requirements diminish, thereby increasing ScvO_2_. In contrast, reduced oxygen delivery to peripheral and splanchnic regions necessitates greater tissue-level oxygen extraction, resulting in lower SfvO_2_ values. Thus, ScvO_2_ and SfvO_2_ cannot be considered interchangeable.

### 4.3. Peripheral Venous Blood

Until a few decades ago, research focused exclusively on the use of central and mixed venous blood. However, with the widespread adoption of pulse oximeters in clinical practice, an increasing number of studies have begun to investigate peripheral venous blood as well [13].

Despite this interest, peripheral venous blood differs from mixed or central venous blood, particularly due to its proximity to the microcirculation, which makes it significantly influenced by local metabolic activity. Recent studies have concluded that peripheral venous blood gas (VBG) analysis cannot replace arterial blood gas (ABG) analysis in patients with respiratory disorders. Nevertheless, Bloom et al. demonstrated that a venous pCO_2_ < 45 mm Hg effectively rules out arterial hypercapnia, with a negative predictive value of approximately 99%.

Therefore, peripheral venous blood can be used as a substitute for arterial blood to assess hemoglobin and electrolytes and for pH and bicarbonate levels—but only in the absence of respiratory disorders [8,14,15,16,17,18]. With regards to lactate, under physiological conditions, peripheral venous lactate levels tend to correlate well with arterial values. However, in pathological states, this concordance is not consistently observed. As the arterial blood composition is independent of the sampling site, it is generally recommended in clinical practice for lactate measurement. Nevertheless, recent studies have demonstrated good agreement between arterial and both central and peripheral venous lactate concentrations [19,20,21,22].

In clinical settings, several parameters are routinely employed to manage patients with cardiovascular compromise, such as those with cardiogenic or septic shock and associated organ dysfunction. However, the literature still lacks consensus on the optimal sample type for evaluating these parameters [16,17]. Currently, arterial and central venous blood are most used, though peripheral venous blood may hold promise as a reliable marker of tissue metabolism due to its proximity to the capillary bed.

## 5. Other Hemodynamic Blood Gas Analytical Indices

### 5.1. Venous Oxygen Saturation

Venous oxygen saturation measured in mixed (SvO_2_) or central (ScvO_2_) venous blood, but not in peripheral venous blood, is considered a surrogate marker of cardiac output [23,24,25,26]. The primary function of the cardiovascular system is to deliver an adequate supply of oxygen to tissues in order to meet the body’s metabolic demands. In cases of cardiovascular impairment, signs of tissue hypoxia may appear and, if prolonged, can progress to MODS and eventually to MOF. Evaluating the adequacy of tissue oxygenation in critically ill patients is therefore essential from both diagnostic and therapeutic standpoints.

The pathophysiological rationale for the clinical use of SvO_2_ is well described by Bloos et al. [27]. Under physiological conditions, arterial oxygen saturation (SaO_2_) is approximately 100%, while SvO_2_ in mixed or central venous blood is around 75%. The remaining 25% represents the oxygen extracted by tissues—termed oxygen extraction ratio (O_2_ER), calculated as SaO_2_ − SvO_2_.

If

CaO_2_ = 1.34 × Hb × SaO_2_;CvO_2_ = 1.34 × Hb × SvO_2_;VO_2_ = CO × (CaO_2_ − CvO_2_).

Then CO can be expressed as: CO = VO_2_/[1.34 × Hb × (SaO_2_ − SvO_2_)].

This equation demonstrates that oxygen consumption (VO_2_) is directly proportional to O_2_ER and therefore inversely proportional to SvO_2_. This is the reason SvO_2_ is used in clinical practice as a surrogate marker of cardiac output. Furthermore, the formula highlights that when cardiac output decreases, O_2_ER increases in an attempt to maintain oxygen consumption (VO_2_). However, O_2_ER can only increase up to a point, beyond which oxygen delivery (DO_2_) falls below the critical threshold, referred to as “critical DO_2_”, leading to anaerobic metabolism and lactate production. The value of critical DO_2_ varies among individuals and is reached more rapidly in frail patients or those with comorbidities [9].

SvO_2_ monitoring is commonly employed to evaluate the effects of various therapies, including mechanical ventilation, inotropic support, blood transfusions, and sedation. Achieving a normal SvO_2_ value (~75%) is often considered a therapeutic target. In the setting of hemodynamic compromise, SvO_2_ values below 75% are associated with a poor prognosis [27].

Unfortunately, while SvO_2_ is highly sensitive, it is not particularly specific. It can be influenced by a range of conditions, including both reduced oxygen delivery (e.g., low cardiac output, hypoxemia, and anemia) and increased oxygen consumption (e.g., fever, pain), even when DO_2_ is preserved. Conversely, SvO_2_ may appear deceptively normal in cases of low cardiac output if microcirculatory dysfunction prevents effective tissue oxygen extraction. This condition, commonly referred to as “cytopathic hypoxia” or “dysoxia,” is thought to be related to endothelial injury, glycocalyx degradation, or microvascular shunting. In such cases, SvO_2_ must be interpreted alongside other markers, particularly lactate levels, for accurate assessment [28].

### 5.2. Relationship Between Lactate and Venous Oxygen Saturation

Tissue hypoperfusion occurs when oxygen demand exceeds the body’s delivery capacity (DO_2_), as seen in sepsis and other critical conditions. Consequently, elevated blood lactate levels are commonly used for the prognostic stratification of critically ill patients, as they suggest a shift toward anaerobic metabolism secondary to inadequate oxygen delivery and result in lactic acid production.

However, lactate elevation is not a specific marker of hypoperfusion. Increased lactate levels may also be observed in the absence of tissue hypoxia, such as in diabetic ketoacidosis (due to an altered glucose metabolism) or during intense physical exertion in untrained individuals. Furthermore, lactate accumulation can occur in liver disease due to impaired hepatic clearance, and in states of elevated adrenergic activity. Catecholamines released during stress stimulate glycolysis, leading to excessive pyruvate production, which is then converted into lactate.

Thus, elevated lactate in septic patients may reflect either an increased adrenergic tone or hypoperfusion associated with organ dysfunction. Interestingly, recent studies have questioned the exclusively negative prognostic value of lactate elevation in sepsis, suggesting that it may also represent an adaptive response. In this view, elevated lactate levels could indicate adrenergic activation in response to a noxious insult, potentially associated with a more favorable prognosis in some cases [29,30].

Like SvO_2_, lactate is a highly sensitive but non-specific parameter. Therefore, their combined interpretation can be particularly valuable in the assessment and stratification of critically ill patients, especially in the context of sepsis. A decrease in SvO_2_, reflecting a reduced cardiac output, does not necessarily correspond to an immediate rise in lactate levels, as lactic acid is only produced when anaerobic metabolism is triggered, i.e., when DO_2_ drops below the so-called critical DO_2_ threshold. Conversely, SvO_2_ values may appear falsely reassuring in the presence of impaired tissue oxygen utilization, such as in the late stages of septic shock, where reduced cardiac output coexists with elevated lactate levels due to a cellular inability to extract or utilize oxygen effectively.

### 5.3. The pCO_2_ Gap: Pathophysiological Basis

In clinical practice, lactate is the most commonly used parameter to assess tissue hypoperfusion, while SvO_2_ is widely employed in intensive care units as a surrogate marker for reduced cardiac output. However, both lactate and SvO_2_ have limitations, as they may yield false positives or negatives. Therefore, the pCO_2_ gap has been introduced and is increasingly utilized to evaluate the hemodynamic status of critically ill patients.

The pCO_2_ gap, also referred to as the venous-arterial (or arterial-venous) pCO_2_ gap or Pv—aCO_2_, reflects the adequacy of the cardiovascular system in clearing CO_2_ produced by tissues. A reduced cardiac output leads to CO_2_ accumulation in the venous blood, resulting in an increased pCO_2_ gap. Several studies have demonstrated that a pCO_2_ gap > 6 mmHg is associated with a worse prognosis and should therefore be considered pathological [31,32]. The pCO_2_ gap is usually calculated using central venous blood, which is more accessible than mixed venous blood. A few studies suggest that peripheral venous blood might also be used [25,26], but current recommendations favor central venous sampling.

CO_2_ is a highly soluble gas generated by energy metabolism under both aerobic and anaerobic conditions. Glycolysis breaks down glucose into two molecules of pyruvate. In the presence of oxygen, pyruvate is converted into acetyl-CoA and enters the Krebs cycle, producing CO_2_. In the absence of oxygen, two types of fermentation occur:Lactic fermentation: Pyruvate is converted into lactate with ATP hydrolysis and CO_2_ production. Lactate may accumulate or be reconverted to pyruvate once aerobic conditions are restored.Alcoholic fermentation: Pyruvate is transformed into ethanol and CO_2_.

Regardless of the metabolic pathway, CO_2_ (or total CO_2_ content, CCO_2_) circulates in the blood in three forms:Bicarbonate ion (HCO_3_^−^): Accounts for 90% of total CO_2_ in arterial blood, formed via the reaction CO_2_ + H_2_O → H_2_CO_3_ → H^+^ + HCO_3_^−^, catalyzed by carbonic anhydrase.Dissolved in plasma: Represented by the partial pressure of CO_2_ (PaCO_2_), approximately 5% of the total CO_2_ in arterial blood.Carbamino compounds: CO_2_ bound to hemoglobin (carbaminohemoglobin), constituting around 1–5% of the total CO_2_ in arterial blood.

Under physiological conditions, the total CO_2_ in arterial blood is approximately 490 mL/L (PaCO_2_ ~40 mmHg), while in venous blood it is about 535 mL/L (PvCO_2_ ~42–45 mmHg). Thus, the normal arterial-venous CO_2_ gap ranges from 2 to 5 mmHg.

Applying Fick’s principle to CO_2_ production (VCO_2_), the arteriovenous pCO_2_ gap can be expressed as a function of cardiac output (CO). Given that

VCO_2_ = CO × (CaCO_2_ − CvCO_2_);PaCO_2_ − PvCO_2_ = K × (CaCO_2_ − CvCO_2_);Then:VCO_2_ = CO × K × (PaCO_2_ − PvCO_2_);Therefore: (PaCO_2_ − PvCO_2_) = VCO_2_/(CO × K).

From this equation, an inverse relationship between CO and the pCO_2_ gap can be deduced. However, the constant K is influenced by several factors, so the relationship is not strictly proportional. Despite these limitations, numerous studies have confirmed that a pCO_2_ gap > 6 mmHg is a reliable hemodynamic marker, particularly in the context of reduced CO [9,32] and a poor prognosis [33].

The relationship between CO and the arteriovenous CO_2_ gap indicates that a marked increase in the pCO_2_ gap occurs only with a significant decrease in CO—typically below 2–3 L/min, and more notably under 1–2 L/min [33,34,35].

Several physiological factors can influence the constant K and affect the pCO_2_ gap.

Haldane Effect: As PaO_2_ increases (e.g., due to oxygen therapy), PaCO_2_ decreases, since oxygenated hemoglobin binds less CO_2_. Conversely, hypoxemia increases the CO_2_ binding to hemoglobin, reducing PaCO_2_ levels.

pH: Changes in bicarbonate (HCO_3_^−^) levels alter the CO_2_ content. In metabolic acidosis, PaCO_2_ tends to be lower, while in alkalosis, it increases.

Hematocrit: Higher hematocrit reduces the plasma volume available for bicarbonate buffering, decreasing PaCO_2_ relative to total CO_2_.

Temperature: Lower temperatures increase CO_2_ solubility, resulting in a higher PaCO_2_ relative to CCO_2_.

These factors shift the CO_2_–CCO_2_ curve to the right under conditions such as hyperoxia, metabolic acidosis, hemoconcentration, and hypothermia, increasing the proportion of dissolved CO_2_ relative to total CO_2_ [9,36].

#### 5.3.1. The pCO_2_ Gap in Clinical Practice

Several studies support the clinical utility of the pCO_2_ gap for the following reasons:It is inversely related to CO in animal models of hemorrhage, hypovolemia, and obstructive shock, like an intermediate-high risk pulmonary embolism [37,38];Persistent elevation after fluid resuscitation in septic shock correlates with a poor prognosis [39];It serves as an early and sensitive marker of impaired venous return [33];It predicts a 28-day mortality in patients with sepsis and organ dysfunction [40];Serial measurements can help monitor therapy effectiveness and guide fluid management when lactate levels remain high but the pCO_2_ gap is normal [38,41].

Thus, combining ScvO_2_, lactate (LAC), and the pCO_2_ gap can enhance diagnostic and therapeutic strategies in hemodynamically unstable patients [42]. For example, elevated lactate (>2 mmol/L) with normal ScvO_2_ (70–75%) and a normal pCO_2_ gap (<6 mmHg) indicates dysoxia—impaired oxygen utilization at the tissue level—rather than circulatory failure, as seen in conditions like anemia.

In sepsis, the pCO_2_ gap may be an earlier indicator of severe hemodynamic compromise than lactate, which only increases when compensatory mechanisms (e.g., O_2_ extraction) are exhausted. A Pv–aCO_2_ gap > 6 mmHg, even with normal lactate, is significantly associated with mortality [43]. Moreover, a persistently elevated Pv–aCO_2_ gap after lactate normalization suggests ongoing hemodynamic dysfunction and predicts poor outcomes. Retrospective studies have also linked elevated postoperative Pv-aCO_2_ gaps to major complications and mortality after cardiac surgery, though diagnostic performance remains limited [44,45].

#### 5.3.2. Limitations of the pCO_2_ Gap

As previously discussed, pCO_2_ is a valuable parameter in hemodynamic monitoring. However, it presents several important limitations. For example: (a) Its interpretation can be misleading in cases of impaired microcirculatory flow or altered CO_2_ production, such as in distributive shock (e.g., anaphylactic shock), where reduced capillary density results in heterogeneous perfusion despite normal cardiac output (CO) and an elevated pCO_2_ gap; (b) as demonstrated by Bakker et al. [46], the pCO_2_ gap in patients with septic shock is primarily influenced by cardiac output and also by the degree of pulmonary dysfunction. Although the pCO_2_ gap is typically larger in non-survivors, its prognostic value remains modest.

Most studies have relied on mixed or central venous blood, and data regarding the clinical utility of peripheral venous samples for pCO_2_ gap measurement are limited. Nonetheless, recent research suggests a good level of agreement between peripheral and mixed venous values [47]. However, to date, no studies have validated the use of the pCO_2_ gap derived from peripheral venous blood for monitoring the hemodynamic status of critically ill patients. Further research is therefore needed before this approach can be routinely implemented in clinical practice.

### 5.4. Ratio of the pCO_2_ Gap to the Arteriovenous O_2_ Content Difference (pCO_2_ Gap/Ca-vO_2_ Gap)

Lactate is a widely used marker of hypoperfusion, reflecting anaerobic metabolism due to reduced oxygen delivery (DO_2_). However, lactate elevation can also result from adrenergic stimulation. This dual origin complicates interpretation in septic patients. Thus, additional tools are needed, such as the ratio between the pCO_2_ gap and the arteriovenous O_2_ content difference (pCO_2_ gap/Ca-vO_2_ gap), which may better reflect anaerobic metabolism [48].

This ratio is derived from the respiratory quotient (RQ = VCO_2_/VO_2_), which is approximately one under normal conditions, as O_2_ consumption matches CO_2_ production. In anaerobic states, VO_2_ decreases disproportionately to VCO_2_, leading to an RQ > 1. A ratio > 1.4 has been associated with anaerobic metabolism in various studies. In practice, elevated lactate combined with a pCO_2_ gap/Ca-vO_2_ gap > 1.4 indicates a true anaerobic metabolism (i.e., DO_2_ deficit), whereas a ratio close to one suggests a lactate increase due to adrenergic stimulation.

These insights may guide therapy, although further studies are needed to determine the ratio’s utility in interpreting lactate dynamics in sepsis and tailoring interventions [49,50].

## 6. Conclusions

Venous blood gas analysis provides not only an alternative but also complementary information to the arterial blood gas analysis. In conclusion, we propose the following recommendations for clinical use:Central venous blood can be used to evaluate ScvO_2_ as a surrogate for SvO_2_ measured from mixed venous blood, with a reasonable agreement of approximately 85–90%, and to assess the pCO_2_ gap in conjunction with the arterial blood gas analysis.Peripheral venous blood is suitable for assessing hemoglobin and electrolytes and for evaluating pH and bicarbonate levels, but only in the absence of respiratory disorders.Arterial blood gas analysis remains the gold standard in critically ill patients for evaluating hemoglobin, electrolytes, lactate, and the acid–base balance, particularly in the presence of respiratory dysfunction.

Table 4 summarizes the practical applications of different blood sample types in critically ill patients.

Peripheral venous lactate shows a good agreement with arterial levels under normal conditions, but this correlation is less consistent in acute illness. Nonetheless, recent evidence suggests central and peripheral venous lactate may be comparable to arterial lactate [19] and a recent review highlights peripheral venous as a reliable marker of metabolic derangements due to its proximity to the capillary exchange [51].

The arteriovenous CO_2_ gap, measured from mixed or central venous blood, is a non-invasive tool for hemodynamic monitoring. It is inversely correlated with CO and can identify early cardiovascular compromise, particularly before DO_2_ becomes critically low and lactate rises. Data on the peripheral venous pCO_2_ gap remain limited [52,53], but it could become a useful and easily accessible tool for assessing hemodynamics in critically ill patients without central venous access.

Given that neither the pCO_2_ gap nor lactate alone reliably reflects anaerobic metabolism or hypoperfusion in all scenarios, the pCO_2_ gap/Ca-vO_2_ gap ratio has emerged as a promising marker of anaerobic conditions in critically ill patients. More research is needed to assess its clinical usefulness in guiding therapy and interpreting lactate trends in sepsis.

## Figures and Tables

**Table 1 medicina-61-01337-t001:** Results of the literature research; VBG = venous blood gas analysis; ED = Emergency Department; ICU = intensive care unit.

	VBG and ED	VBG and CRITICAL CARE	VBG and ICU
Abstract	180	308	112
Review	7	28	5
Systematic review	0	2	0
Systematic review and meta-analysis	2	2	0
Clinical trial	11	23	9

**Table 2 medicina-61-01337-t002:** Difference between arterial and venous blood of pH, pCO_2_, HCO_3_^−^, and pO_2_, respectively, indicated as Δ(a-v) pH, Δ(a-v) pCO_2_, Δ(a-v) HCO_3_^−^, and Δ(a-v) pO_2_.

Δ(a-v) pH	~0.04 units
Δ(a-v) pCO_2_	~6 mm Hg
Δ(a-v) HCO_3_^−^	~1 mEq/L
Δ(a-v) pO_2_	~55 mmHg

**Table 3 medicina-61-01337-t003:** Difference between arterial and mixed, central, and peripheral blood gas analysis in normal conditions.

Parameter	Arterial	Mixed Venous	Central Venous	Peripheral Venous
pH	7.40 (7.35–7.45)	7.36 (7.35–7.45)	7.35–7.37	7.36 (7.29–7.45)
pO_2_	80–100 mmHg	35–40 mmHg	35–40 mmHg	35–40 mmHg
SO_2_	95–100%	~75%	70–75% (lower due to cardiac and cerebral oxygen consumption)	65–75% (variable)
pCO_2_	35–45 mmHg	41–51 mmHg	44–45 mmHg (range: 35–48)	43–48 mmHg
HCO_3_^−^	22–26 mmol/L	23–29 mmol/L	25 mmol/L (range: 22–27)	25–26 mmol/L
Electrolytes, Hemoglobin	Identical values	Identical values	Identical values	Identical values
Lactate	0.5–1.8 mmol/L	Comparable under normal conditions; agreement is controversial in pathological states	Similar to mixed venous	Similar to mixed venous

**Table 4 medicina-61-01337-t004:** Practical use of different blood samples in critically ill patients. Details are extensively discussed in the text. ✔: reliable; ✖: not reliable.

	Arterial Blood	Mixed Venous Blood	Central Venous Blood	Peripheral Venous Blood
pH	✔	✔	✔	✔
pO_2_	✔	✖	✖	✖
SO_2_	✔	✔ (useful for venous oxygen saturation)	✔ (useful for venous oxygen saturation)	✖
pCO_2_	✔	✖ (but useful for pCO_2_ gap)	✖ (but useful for pCO_2_ gap)	✖
HCO_3_^−^	✔	✔	✔	✔
Electrolytes, Hemoglobin	✔	✔	✔	✔
Lactate	✔	✔ (only in normal conditions)	✔ (only in normal conditions)	✔ (only in normal conditions)

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
