# Peer review of "The Role of Venous Blood Gas Analysis in Critical Care: A Narrative Review"

_medicina, 2025, doi:10.3390/medicina61081337_

Round 1
Reviewer 1 Report
Comments and Suggestions for Authors
The authors report a narrative review after conducting a medical literature search for the past 10 years. They found 89 studies and article section was done by two reviewers. They included meta-analysis, systematic reviews and clinical trials. They report venous pH, CO2 and HCO3 as closer to arterial values, especially pH and HCO3. They report the values from different studies in mixed venous and peripheral venous as well. The pCO2 gap/AV O2 content difference was also assessed.
The findings are relevant. The manuscript is well-written.
I have minor suggestions:
- Reports the dates of the literature search, or at least up to what date was the literature search done.
- Report number of abstracts found and review.
Author Response
- Reports the dates of the literature search, or at least up to what date was the literature search done.
We reported the dates of the literature search, as suggested.
- Report number of abstracts found and review.
We reported number of abstracts found and review, as suggested.
Reviewer 2 Report
Comments and Suggestions for Authors
General Comments
This is a well-structured and informative narrative review on the clinical use of venous blood gas (VBG) analysis in acutely ill patients. The manuscript walks the reader through the physiological, technical and clinical nuances of VBG, comparing it to the more established arterial blood gas (ABG) measurements. The authors bring together a substantial amount of evidence and integrate it with physiological reasoning in a way that is accessible yet thorough.
However, despite its strengths, the manuscript leans heavily on exposition and, at times, feels like a well-annotated lecture rather than a critical review. The writing is clear but occasionally too neutral—there are moments where the authors could and should take a position. For instance, the evidence supporting the use of peripheral venous pCOâ‚‚ as a rule-out tool for hypercapnia is good enough to be stated with more confidence. Conversely, the practical limitations of the pCOâ‚‚ gap in real-time decision-making deserve a sharper critique.
Specific Comments
.- Introduction. It is succinct and establishes the problem well: ABGs are invasive and not without risks; VBGs offer a tempting alternative. It would help to articulate more clearly the aim of the review. Is the intention to defend VBG as a substitute? Or rather, to outline when and how it can safely complement ABG? That distinction matters.
.- Methods. The search strategy is described in general terms, with major databases and relevant keywords included. However, there is no mention of whether study quality was assessed or how data were synthesized. The manuscript draws conclusions based on existing studies, it would be reassuring to know how the authors weighed evidence or dealt with heterogeneity.
.- Results, ok.
.- Tables. Table 3, summarizing differences across sampling sites, is practical and should be retained in any final version.
However, Table 1, listing the number of reviews and clinical trials found, is less informative unless paired with an evaluation of their quality or clinical relevance.
.- Discussion. The physiology is well explained. The sections on oxygen extraction, central vs. peripheral venous sampling, and the interpretation of SvOâ‚‚ and ScvOâ‚‚ are both scientifically solid and clinically sound. The explanation of why ScvOâ‚‚ can be misleading in septic shock is particularly well handled.
The discussion on lactate and its limitations is fair, and the mention of "stress hyperlactatemia" is an important one. The explanation of the pCOâ‚‚ gap and its theoretical underpinnings is accurate and helpful, though again, one wishes the authors would venture a bit more into the messy reality of clinical practice: these measurements are often delayed, inconsistent, or poorly interpreted.
The concept of tissue oxygenation and the limits of surrogate markers is revisited multiple times. Please, review.
.- Conclusions are cautious and appropriately.
.- References. 49 quotes, of which approximately only 10 (just over 20%) have been published in the last five years. The reader would appreciate including any additional recent quotes. The citation style is correct, and the sources are well chosen.
Author Response
- …However, despite its strengths, the manuscript leans heavily on exposition and, at times, feels like a well-annotated lecture rather than a critical review. The writing is clear but occasionally too neutral—there are moments where the authors could and should take a position. For instance, the evidence supporting the use of peripheral venous pCOâ‚‚ as a rule-out tool for hypercapnia is good enough to be stated with more confidence. Conversely, the practical limitations of the pCOâ‚‚ gap in real-time decision-making deserve a sharper critique.
As suggested, we tried to take position when possible. For example, we focused on the use of peripheral venous pCOâ‚‚ as a rule-out tool for hypercapnia and we highlighted the limitations associated with the use of the pCO2 gap as a diagnostic parameter.
- It is succinct and establishes the problem well: ABGs are invasive and not without risks; VBGs offer a tempting alternative. It would help to articulate more clearly the aim of the review. Is the intention to defend VBG as a substitute? Or rather, to outline when and how it can safely complement ABG? That distinction matters.
As suggested, we tried to articulate more clearly the aim of the review.
- The search strategy is described in general terms, with major databases and relevant keywords included. However, there is no mention of whether study quality was assessed or how data were synthesized. The manuscript draws conclusions based on existing studies, it would be reassuring to know how the authors weighed evidence or dealt with heterogeneity. Table 1, listing the number of reviews and clinical trials found, is less informative unless paired with an evaluation of their quality or clinical relevance.
We did not assess the quality of the studies using validated tools, due to the narrative nature of this review, which explores various aspects of the potential clinical utility of venous blood. In addition to systematic reviews and meta-analyses, we prioritized clinical studies with robust methodological designs, large sample sizes, and, when possible, more recent publication dates.
- The concept of tissue oxygenation and the limits of surrogate markers is revisited multiple times. Please, review.
As suggested, we reviewed the text, avoiding to revisit the concept multiple times.
- 49 quotes, of which approximately only 10 (just over 20%) have been published in the last five years. The reader would appreciate including any additional recent quotes. The citation style is correct, and the sources are well chosen.
As suggested, we added some recent papers, as the following:
Eugene Yuriditsky, Robert S Zhang, Jan Bakker, James M Horowitz, Peter Zhang, Samuel Bernard, Allison A Greco, Radu Postelnicu, Vikramjit Mukherjee, Kerry Hena, Lindsay Elbaum, Carlos L Alviar, Norma M Keller, Sripal Bangalore, Relationship between the mixed venous-to arterial carbon dioxide gradient and the cardiac index in acute pulmonary embolism, European Heart Journal. Acute Cardiovascular Care, Volume 13, Issue 6, June 2024, Pages 493– 500, https://doi.org/10.1093/ehjacc/zuae031.
Nassar, B., Badr, M., Van Grunderbeeck, N. et al. Central venous-to-arterial PCO2 difference as a marker to identify fluid responsiveness in septic shock. Sci Rep 11, 17256 (2021). https://doi.org/10.1038/s41598-021- 96806-6.
Bijapur MB, Kudligi NA, Asma S. Central Venous Blood Gas Analysis: An Alternative to Arterial Blood Gas Analysis for pH, PCO2, Bicarbonate, Sodium, Potassium and Chloride in the Intensive Care Unit Patients. Indian J Crit Care Med. 2019 Jun;23(6):258-262. doi: 10.5005/jp-journals-10071-23176. PMID: 31435143; PMCID: PMC6698350.
Reviewer 3 Report
Comments and Suggestions for Authors
Please find the attached file

Author Response
- Line 45. Accidental puncture...is there a better way to say it?... needlestick injury to the
We corrected the mistake, as suggested.
- Literature search
We corrected the mistake, as suggested.
- Differences in the main gas analysis parameters of arterial and venous blood
Line 76...influenced by the local…
We corrected the mistake, as suggested.
- Mixed, Central and Peripheral Venous Blood
4.1 Mixed venous Lines 116-128...There are many information on the SvO2, oxygen extraction, but there are no reference cited...These numbers are well known, but we need a citation.
- We added some references, as suggested:
Janotka M, Ostadal P. Biochemical markers for clinical monitoring of tissue perfusion. Mol Cell Biochem. 2021 Mar;476(3):1313-1326. doi: 10.1007/s11010-020-04019-8. Epub 2021 Jan 2. PMID: 33387216; PMCID: PMC7921020.
Marti YN, de Freitas FG, de Azevedo RP, Leão M, Bafi AT, Machado FR. Is venous blood drawn from femoral access adequate to estimate the central venous oxygen saturation and arterial lactate levels in critically ill patients? Rev Bras Ter Intensiva. 2015 Oct-Dec;27(4):340-6. doi: 10.5935/0103-507X.20150058. PMID: 26761471; PMCID: PMC4738819.
- Lines 129-146...Again, a lot of information, but without a cited reference. All this information here seems too much to comprehend. This section should be rewritten...What is the importance of the findings of the mentioned studies...apart from the numbers, there are not much information that a clinician could use... So, yes, there are differences, but what do they mean in practice....
We added references, as suggested. We modified this section, however we believe that it is important to know these parameters to identify the appropriate clinical context for their use and how to combine them effectively.
- Lines 151,152 The subsection should be numerated 4.2 Central venous...not 3.2, the same stands for peripheral venous blood
We corrected the mistake, as suggested.
- Lines 154-162...Again, no references cited and a lot of information... Line 172........18 %...THE CITATION???
We added the references, as suggested.
- Line 188. Incorrectly numerated Peripheral venous blood. Line 200...Firstly...Line 202...Secondly....
We corrected the mistakes, as suggested.
- The section about peripheral venous blood is not clear to me...please state clearly why are there differences between central venous and peripheral venous blood...cite the studies in which peripheral venous blood gasses were used for patient monitoring...the section should be rewritten
We rewrote this section, emphasizing the clinical contexts in which peripheral venous blood should be used and differences between peripheral and central venous blood, including the relevant references.
- 5. Other hemodynamic blood gas analytical indices. Line 229…Incorrectly numerated subsection Line 282-288.
We corrected the mistake, as suggested.
- Again NO REFERENCE CITED??????
We added the references, as suggested.
- 6. Conclusion It should be, MAYBE, rewritten in the sense to address directly when one should use central venous blood gasses, peripheral venous blood gasses ....
We rewrote the conclusions in order to address directly when one should use central venous blood gasses, peripheral venous blood gasses.
- Some reference that could be included, please consider
Eugene Yuriditsky, Robert S Zhang, Jan Bakker, James M Horowitz, Peter Zhang, Samuel Bernard, Allison A Greco, Radu Postelnicu, Vikramjit Mukherjee, Kerry Hena, Lindsay Elbaum, Carlos L Alviar, Norma M Keller, Sripal Bangalore, Relationship between the mixed venous-to-arterial carbon dioxide gradient and the cardiac index in acute pulmonary embolism, European Heart Journal. Acute Cardiovascular Care, Volume 13, Issue 6, June 2024, Pages 493– 500, https://doi.org/10.1093/ehjacc/zuae031.
Nassar, B., Badr, M., Van Grunderbeeck, N. et al. Central venous-to-arterial PCO2 difference as a marker to identify fluid responsiveness in septic shock. Sci Rep 11, 17256 (2021). https://doi.org/10.1038/s41598-021- 96806-6.
- We added the suggested references.
Reviewer 4 Report
Comments and Suggestions for Authors
I have read with great interest this narrative review by Dario G et al concerning the role of venous blood gas analysis in critical care. Overall the paper is well written and comprehensive. It summarizes adequately the information that can be derived by the gas analysis of a venous sample in the emergency and critical care medicine. In these clinical settings there is a constant need for practical, easily accessible and complementary sources of data concerning the acid base balance and the hemodynamic status of the patient. I have no major comments. Some minor suggestions would include:
- A more critical discussion of the available information would be desirable
- A graphical representation of key messages would be appealing
- Some information concerning SvO2 are repeated in different paragraphs
Author Response
- A more critical discussion of the available information would be desirable
We tried to add a more critical discussion of the available information.
- A graphical representation of key messages would be appealing
Tables 4 summarizes the practical use of different samples blood in critical ill patients.
- Some information concerning SvO2 are repeated in different paragraphs
As suggested, we reviewed the text, avoiding to repeat information concerning SvO2 multiple times, in particular paragraph 5.1.
Round 2
Reviewer 3 Report
Comments and Suggestions for Authors
It is my opinion that you made substantial changes to the manuscript making it suitable for publication, I am looking forward to reading the article.....